# Unmasking the Complex Interplay of Obesity Hypoventilation Syndrome, Heart Failure, and Sleep Dysfunction: A Physiological and Psychological Perspective in a Digital Health World

**DOI:** 10.3390/bs15030285

**Published:** 2025-02-28

**Authors:** Elvia Battaglia, Valentina Poletti, Elena Compalati, Matteo Azzollini, Eleonora Volpato

**Affiliations:** 1IRCCS Fondazione Don Carlo Gnocchi, 20148 Milan, Italy; ebattaglia@dongnocchi.it (E.B.); valentina.poletti@unicatt.it (V.P.); ecompalati@dongnocchi.it (E.C.); 2Sleep Center, Centro Diagnostico Italiano—C.D.I., 20147 Milan, Italy; matteo.azzollini@cdi.it; 3Dipartimento di Psicologia, Università Cattolica del Sacro Cuore, 20123 Milan, Italy; 4Research Group Health Psychology, University of Leuven, 3000 Leuven, Belgium

**Keywords:** obesity hypoventilation syndrome, heart failure, sleep dysfunction, digital health, narrative review

## Abstract

Obesity hypoventilation syndrome (OHS) is a multifaceted condition characterized by significant respiratory, cardiovascular, and psychological consequences. Positive airway pressure (PAP) therapy remains the cornerstone treatment, improving respiratory function, neurocognition, and mental health disorders such as depression and anxiety. However, its long-term impact on quality of life, physical activity, and broader health outcomes is not fully understood. Challenges such as residual apnoea/hypopnea index, reduced physical activity, and impaired quality of life persist despite high adherence rates. Factors like hypercapnia and daytime respiratory symptoms play a pivotal role in patient outcomes, underscoring the need for strategies beyond adherence alone. This review explores the interplay between OHS, heart failure, and sleep dysfunction, advocating for personalized PAP settings, targeted management of residual respiratory events, and enhanced patient education. Digital health technologies, including remote monitoring and feedback systems, present promising tools to optimize care delivery and foster holistic management. By integrating physiological, psychological, and digital health perspectives, this narrative review aims to advance understanding and improve outcomes for patients with OHS and other complex sleep-disordered breathing conditions.

## 1. Introduction: Bridging Complexities in Obesity Hypoventilation Syndrome, Heart Failure, and Sleep Disorders

Hypoventilation syndromes are a group of illnesses that collectively produce alveolar hypoventilation. Insufficient ventilation resulting in hypercapnia, or a rise in the partial pressure of carbon dioxide as determined by arterial blood gas measurement (PaCO_2_), is known as alveolar hypoventilation ([76]; [1]). Six categories of hypoventilation disorders are listed in the section on sleep-related respiratory disorders of the International Classification of Sleep Disorders: obesity hypoventilation syndrome (OHS), idiopathic alveolar hypoventilation, congenital alveolar hypoventilation brought on by late hypothalamic dysfunction, hypoventilation during sleep linked to drugs or treatments, and hypoventilation during sleep brought on by various illnesses ([36]).

Sleep-disordered breathing (SDB), obesity (body mass index (BMI) ≥ 30 kg/m^2^), and daytime hypercapnia (partial pressure of carbon dioxide (PaCO_2_) ≥ 45 mmHg) are the three hallmarks of OHS when there are no other neuromuscular, mechanical, or metabolic reasons for hypoventilation ([97]). OHS affects approximately 0.3–0.4% of the general population and 8–20% of the patients with obesity ([11]; [97]). Its prevalence varies widely between studies, partly due to differences in sample characteristics, diverse disease definitions, and heterogeneity in assessment methods ([12]). OHS is often under-recognized, with a delayed diagnosis, typically occurring in the fifth or sixth decades of life when patients present with acute-on-chronic hypercapnic respiratory failure or are evaluated by respiratory or sleep specialists. Nearly 75% of patients with OHS have been misdiagnosed and treated for obstructive lung disease (most commonly chronic obstructive pulmonary disease (COPD)), despite having no evidence of obstructive physiology on pulmonary function testing ([55]; [61]). Furthermore, almost 90% have obstructive sleep apnoea (OSA), characterized by an apnoea/hypopnea index (AHI) ≥ 5 events/h, and around 70% of people with OHS present with severe OSA (AHI ≥ 30 events/h). Non-obstructive sleep hypoventilation with moderate or non-existent OSA is present in the remaining patients ([55]).

The focus of this narrative review will be on the conjunction between obesity hypoventilation syndrome (OHS), heart failure (HF), and sleep disorders (SD), which together represent a challenging triad with intricate interconnections, significantly impacting patients’ physical, cognitive, and psychological well-being ([21]). OHS, HF, and SD conditions often coexist, creating a cycle of daytime hypoventilation, SD, and cardiovascular strain, which exacerbates functional impairments and quality-of-life reductions ([90]; [98]). Despite advancements in therapeutic approaches, such as non-invasive ventilation (NIV) and positive airway pressure (PAP) therapies, research remains limited in evaluating comprehensive interventions targeting both physiological and psychological aspects of care. Bridging these complexities requires innovative strategies that integrate personalized therapies, patient education, and emerging digital health technologies to improve outcomes and optimize long-term management.

### Aim of the Review

The primary objective of this narrative review is to provide an updated synthesis of the evidence on the interplay between OHS, HF, and SD. Specifically, it seeks to explore how these conditions interact pathophysiologically and psychologically and to identify the role of emerging technologies in characterizing and monitoring the associations between sleep dysfunction and cardiovascular outcomes. This review also aims to highlight differences in diagnostic and therapeutic approaches enabled by advancements in digital health tools.

## 2. Methods

A narrative or nonsystematic review is a complete narrative synthesis of previously published material that gives a wide viewpoint on a topic ([10]; [29]). This narrative review has been structured considering the Scale for the Assessment of Narrative Review Articles (SANRA), a brief critical appraisal tool for the assessment of non-systematic articles ([10]).

### 2.1. Search Strategy

Studies were identified by searching the electronic databases Medline, Embase, PsychINFO, and Cinhal, from inception up to December 2024. The search strategy was originally developed for Medline and translated into other databases using the appropriate Boolean operators, filters, and controlled vocabulary as applicable. The MeSH Terms used to search all databases were as follows: obesity hypoventilation syndrome (OHS), heart failure (HF), sleep disorders (SD), sleep disorder breathing (SDB), obstructive sleep apnoea (OSA), physiology, pathophysiology, psychological well-being, anxiety, depression, digital therapies (DTx), digital health innovations (DHIs), decision aids (DAs), artificial intelligence (AI), and shared decision making (SDM).

### 2.2. Study Selection Criteria

Table 1 shows the main inclusion and exclusion criteria of this narrative review. Only English-language articles were considered as eligible. To find additional pertinent publications, the bibliographies of the primary articles (reviews, systematic reviews, and meta-analyses) were also examined.

### 2.3. Data Extraction and Synthesis Methods

Considering the complexity involved in describing the intersection between OHS, HF, and SDB from both physiological and psychological perspectives, as well as the considerable heterogeneity of interventions classified as digital, this review was not conceived as a systematic literature review. Instead, it adopts a narrative approach, allowing for a comprehensive and flexible synthesis of the available literature. Given the variety of outcome measures reported across studies, a formal data extraction protocol typically employed in systematic reviews was not implemented. Rather, key themes and findings were extracted and synthesized based on their relevance to the research question, aiming to provide clinicians with a useful overview of the current state of knowledge on the interplay between OHS, HF, and SDB (Table 2). This approach enables a more complete view while accommodating the diverse nature of the studies reviewed.

## 3. Results

From the literature review, the main themes emerging from the research question are as follows: the pathophysiology and mechanistic aspects of OHS, HF, and SDB, along with their intersections; psychosocial and cognitive aspects; potential digital health interventions; and therapeutic approaches.

### 3.1. Pathophysiology and Mechanistic Insights into OHS, HF, and SDB

#### 3.1.1. Definition and Diagnosis of Sleep Hypoventilation

The American Academy of Sleep Medicine (AASM) defines sleep hypoventilation as a PaCO_2_ (or surrogate) >55 mmHg for more than 10 min or an increase in PaCO_2_ > 10 mmHg compared to awake supine values, exceeding 50 mmHg for more than 10 min ([14]). Nocturnal polysomnography or respiratory polygraphy is essential to identify patterns of sleep-disordered breathing (SDB), including obstructive and non-obstructive hypoventilation, to tailor therapy effectively, especially for mechanical ventilation.

#### 3.1.2. Pathophysiology of Obesity Hypoventilation Syndrome (OHS)

Many studies in the literature have shown that body mass index (BMI) is a risk factor for left ventricular (LV) remodeling, such as LV hypertrophy and dilatation, which can lead to heart failure (HF) ([62]). In order to maintain eucapnia and compensate for hypercapnia, obese patients increase their respiratory effort. When this drive is compromised, hypoventilation first manifests during rapid eye movement (REM) sleep and develops as a result of a confluence of factors such as decreased central respiratory drive, compromised diaphragm function, and mechanical limitations brought on by obesity. This repetitive hypoventilation leads to secondary depression of respiratory centers, resulting in daytime hypercapnia and OHS ([73]). The significant occurrence of central hypoventilation during REM sleep in OHS results from leptin resistance. Leptin, an effective respiratory enhancer, loses efficacy in OHS, which disrupts respiratory regulation and leads to cardiometabolic issues. In 2012, the European Respiratory Society (ERS) proposed a severity grading for OHS, ranging from early stages with elevated bicarbonate levels or sleep hypoventilation to advanced stages with daytime hypercapnia and significant comorbidities ([14]).

The pathophysiology of OHS is related to three main mechanisms ([55]):*Obesity-related changes in the respiratory system*: excess adipose tissue in the abdomen and chest wall reduces lung compliance, impairs diaphragmatic motion, and increases airway resistance, necessitating a higher respiratory drive;*Changes in respiratory drive:* a failure to sustain increased respiratory drive results in hypoventilation;*Sleep-related breathing abnormalities*: Obesity-associated physiological changes, such as excessive fat deposition around the upper airway and reduced lung volumes, increase pharyngeal collapsibility during supine sleep ([49]).

Fluid overload and lower extremity edema, particularly in uncompensated acute cardiogenic or respiratory conditions, may lead to a nocturnal rostral fluid shift ([70]). This shift contributes to upper airways narrowing and obstructive events during sleep. The amount of fluid moving from the legs to the upper body during the night correlates with sleep apnea severity, and preventing nocturnal fluid shifts may reduce this severity ([50]; [70]). Identifying the dominant mechanisms in a patient is crucial to determining their OHS phenotype and predicting responses to positive airway pressure (PAP) therapy ([69]).

### 3.2. Systemic Complications of OHS

OHS is associated with several systemic complications, including the following:*Arterial hypertension* (prevalence 55–88%) ([16]);*Metabolic and cardiovascular diseases* ([74]);*Pulmonary hypertension*, related to chronic hypoventilation. About half of patients with OHS have pulmonary hypertension ([3]);*Cardiac arrhythmias and atrial fibrillation* (AF), associated with increased obesity. Several longitudinal studies have demonstrated an association between obesity and an increased risk of sudden cardiac death (SCD) ([67]; [85]);*Heart failure*, with obesity contributing to adverse haemodynamic changes, cardiac remodeling, and ventricular dysfunction ([5]) (Figure 1).

### 3.3. Heart Failure in Obesity

#### 3.3.1. Pathophysiological Mechanisms

These systemic complications are most pronounced in severely obese individuals and may predispose to the development of HF, even in the absence of comorbidities such as coronary artery disease (CAD), valvular heart disease, pericardial disease, and congenital heart disease ([47]).

The Framingham Heart Study ([52]) showed that HF had developed in 8.4% of their study population and that the risk of developing heart failure was approximately doubled in people with obesity (BMI 30 kg/m^2^ or greater) compared with the non-obese population ([44]). Based on their findings, they estimated that about 11% of HF cases in men and 14% of HF cases in women could be secondary to obesity alone. In obese patients, higher body mass index is associated with higher total blood volume and cardiac output. The increase in cardiac output and cardiac work is due to an increase in LV stroke volume and stroke work ([4]). This increase was originally attributed to increased fat mass, but more recent studies suggest that in class I and class II obesity, it is predominantly due to an increase in fat-free mass ([6]). Obesity reduces peripheral vascular resistance (PVR), increasing cardiac output, which can reach up to 10 L/min in severe cases ([6]).

Autopsy studies on class III obesity reveal increased heart weight, LV wall thickness, microscopic LV hypertrophy (LVH), variable RV wall thickening, and excess epicardial fat ([4]).

Left ventricular internal dimension during diastole, wall thickness, and mass are greater in obese patients than in lean patients ([5]); the duration of obesity also appears to be an important factor in its development ([47]), and obesity itself would predispose a patient to left ventricular diastolic dysfunction, which occurs in 12% of class I, 35% of class II, 45% of class III obese patients ([66]).

#### 3.3.2. Heart Failure in Obesity: Clinical Findings

A clinical study performed by Mohammad Reza and colleagues ([62]) suggests that diastolic dysfunction may partly explain the increased incidence of heart failure in the present study population.

Obesity is associated with several changes in cardiovascular function, which are related to an increase in cardiac parameters such as LV dimensions, stroke volume, cardiac output, and eccentric hypertrophy. All these parameters can lead to diastolic heart failure.

Arterial hypertension occurs in up to 50% of obese patients ([4]), and the effect of this coexisting condition on LV morphology depends on the relative severity and duration of both conditions. Arterial hypertension alone promotes the development of concentric or eccentric remodeling of left ventricular hypertrophy ([96]).

Potential reasons for anatomical modifications in obese patients include the severity and duration of arterial hypertension, activation of the sympathetic nervous system, the RAAS, and the effects of growth factors, such as insulin ([4]; [47]). Metabolic abnormalities contributing to LV systolic and diastolic dysfunction or hypertrophy have been identified in animal models, including lipocyte apoptosis, insulin resistance with hyperinsulinemia, leptin resistance with hyperleptinemia, reduced adiponectin levels, and activation of the RAAS ([44]).

#### 3.3.3. Obesity Paradox in Heart Failure

Despite the strong association between obesity, cardiovascular risk factors, and cardiovascular diseases, studies on HF in obese patients reveal an “obesity paradox”, where obese patients with cardiovascular diseases often exhibit a more favorable prognosis compared to normal-weight or underweight patients ([48]; [56]).

Horwich et al. observed the best heart failure outcomes in overweight patients, followed by obese patients, while the worst outcomes occurred in underweight patients, closely followed by those with normal BMI ([32]).

The reasons for this paradox remain unclear but may involve increased catabolic states in lower-weight individuals and protective neuroendocrine profiles in obesity ([7]). Adipose tissue in obese patients produces tumor necrosis factor-α (TNF-α) receptors, neutralizing TNF-α’s adverse effects in acute and chronic heart failure. Certain neurohormonal and metabolic derangements in obesity, such as increased sympathetic tone, RAAS activation, hyperinsulinemia, hyperleptinemia, and reduced adiponectin, may contribute to LV hypertrophy and better prognosis ([9]; [20]). Additionally, an attenuated RAAS response and higher circulating lipoproteins in obese patients may bind lipopolysaccharides, mitigating inflammatory cytokine effects ([40]; [59]).

Myocardial lipotoxicity, where excess fatty acids and triglycerides accumulate in cardiomyocytes, leads to dysfunction, remodeling, and fibrosis. These processes are mediated by long-chain fatty acids and their metabolites, though their relevance to humans remains uncertain. Obese patients also show elevated myocardial triglycerides, correlating with LV hypertrophy and dysfunction. Lower levels of natriuretic peptides (BNP, NT-proBNP) in obese patients may reflect increased muscle mass, while higher fat mass is associated with greater muscle strength ([2]; [95]).

Lastly, untreated OHS is linked to significant comorbidities and high mortality, with observational studies reporting a 24% all-cause mortality at 1.5–2 years ([2]; [58]).

## 4. Psychosocial and Cognitive Dimensions of OHS and Related Respiratory Conditions

Although the psychological and cognitive characteristics of patients with OHS, HF and SDB have been less extensively studied, existing research highlights important associations. OHS is linked to poor sleep quality, reduced quality of life ([2]; [58]) and excessive daytime sleepiness, all of which contribute to impaired cognitive performance ([42]; [89]) as well as the raise of psychological distress, anxiety and/or depression ([8]). Moreover, severe SDB is strongly associated with hypoventilation resulting from obesity, exacerbating hypoxaemia and sleep fragmentation—two key factors that may serve as precursors to cognitive difficulties related to memory, attention or exective functions in patients with OSA ([89]). Animal and human studies on hypercapnia provide further insight: in both cases, exposure to elevated carbon dioxide levels led to reduced speed and amplitude of neural oscillations, as measured by electroencephalograms (EEGs) ([26]; [93]). In humans, hypercapnia also impaired mental and psychomotor functions. While these findings suggest that hypercapnia can induce significant but reversible changes in brain electrical activity, its role in cognitive impairment among patients with OSA and hypoventilation remains unclear ([31]). One small study involving obese patients with OSA found negative correlations between awake hypercapnia and cognitive measures, such as memory, processing speed, and reaction time. However, the absence of transcutaneous carbon dioxide (TcCO_2_) monitoring during sleep limited the ability to determine whether isolated sleep hypoventilation—often considered a prodromal stage of OHS—also contributes to cognitive deficits ([46]). Another study examining patients with OHS and severe OSA found a negative association between EEG slowing during non-rapid eye movement (NREM) sleep and psychomotor vigilance. However, this study also did not investigate the relationship between sleep-associated TcCO_2_ levels and cognitive performance ([78]).

Home mechanical ventilation (HMV) has been shown to significantly improve the quality of life (QoL) in patients with chronic respiratory failure, yet it remains unclear which patient subgroups derive the most benefit. A prospective study assessed QoL changes over six months in 74 patients who initiated HMV, using the Severe Respiratory Insufficiency Questionnaire. The findings revealed that patients with severe baseline impairment, those requiring oxygen supplementation, and those who started ventilation in an acute setting experienced the greatest improvements. Diagnosis was also a critical factor: patients with COPD and obesity hypoventilation syndrome showed the most substantial QoL gains, while individuals with ALS did not demonstrate notable improvements ([88]). These results underscore the transformative potential of HMV, particularly for specific patient profiles, while showing that the mode of ventilation does not influence outcomes. Importantly, the study also highlights how chronic respiratory conditions, often accompanied by cognitive difficulties, may exacerbate psychological distress. Impaired cognitive functioning—linked to factors such as sleep disturbances, hypoxaemia, and hypercapnia—can affect memory, attention, and psychomotor skills, potentially fostering feelings of frustration, anxiety, and helplessness. By improving respiratory function, sleep quality, and related symptoms, HMV may help alleviate cognitive difficulties and, in turn, reduce the psychological distress associated with these conditions, further enhancing overall QoL.

Further research is needed to elucidate the complex interplay between hypoventilation, hypercapnia, sleep disturbances, and both cognitive function and quality of life in patients with OHS and related conditions.

## 5. Digital Health Solutions: Transforming Management of OHS, HF, and SDB

Given the complex interplay between OHS, HF, and SDB, as well as the substantial socio-economic burden they impose, significant efforts have been made in recent years to develop methodologies that leverage the Internet and digital technologies (DTx) ([15]). These advancements have facilitated the implementation of telemedicine interventions, supported by state-of-the-art tools ([77]). The broader scope of digital health now encompasses teleconsultations, telemonitoring, smartphone applications, noninvasive remote monitoring devices, wearables, implantable devices, and sensors ([25]). Emerging fields such as advanced computing sciences, big data analytics, genomics for personalized medicine, and artificial intelligence are also increasingly recognized as digital health innovations (DHIs) ([77]). These tools are being progressively adopted to enhance access to healthcare services, reduce costs and waiting times, and improve the quality and personalization of interventions. In conditions like OHS, HF, and sleep disorders, DHIs offer the potential to prevent disease progression and reduce healthcare expenditures. Enhanced monitoring capabilities facilitate early detection of clinical deterioration and timely interventions, thus, improving patient outcomes. Teleconsultation and remote monitoring, in particular, help reduce unnecessary hospitalizations, enable continuous disease surveillance, support efficient management strategies, and ultimately improve clinical outcomes. ([30]).

The current body of research on the interplay between OHS, HF, sleep disturbances, and the application of intervention and monitoring technologies remains limited. While the advantages of DHIs are evident, several barriers exist at individual, system, and policy levels that merit careful consideration.

At the individual level, clinicians demonstrate mixed attitudes toward the adoption of DHIs. While some remain skeptical about their effectiveness, others view them as valuable tools for enhancing care delivery. The growing prevalence of remote consultations, such as video consultations, has revealed technical challenges; however, these are expected to diminish as technological advancements continue. A critical challenge is the integration of decision aids (DAs) into electronic medical records, which is perceived as a complex but essential step for facilitating shared decision making (SDM). Streamlining the incorporation of DAs into existing care pathways and ICT systems could address the frequently cited barrier of limited consultation time, ultimately improving workflow efficiency ([79]).

On the patient side, preferences and capabilities vary significantly. Older adults ([68]) and individuals with low health literacy ([94]) often encounter difficulties using digital tools, underscoring the importance of equitable access when developing DAs. To address this, patient preferences for paper-based or digital DAs should be considered. While digital formats provide comprehensive and detailed information, paper-based options offer succinct and accessible content, which may be preferred by certain populations. Importantly, no definitive evidence currently supports the superiority of one format over the other. Thus, the selection of DA format should be guided by patient needs, clinician perceptions, and available healthcare resources.

Overall, ensuring that DHIs are adaptable to diverse patient populations and seamlessly integrated into clinical workflows is essential for maximizing their potential to improve care outcomes and promote equity.

Finally, as emphasized by the latest recommendations from international guidelines, such as those of the American College of Cardiology ([30]) and the European Society of Cardiology ([57]), the rapid evolution of DHIs poses challenges for regulators, reimbursement authorities, and medical professionals in evaluating their value and integration into routine care. Moreover, the low personal cost of training in the use of big data analysis and digital control tools for workplace monitoring and assessment is one of the cultural and structural barriers that exist even if digital transformation lowers workplace hazards.

## 6. Therapeutic Approaches: Mechanical Ventilation and Integrated Interventions

Currently, there is no standardized protocol for managing OHS. Treatment focuses on correcting SDB, weight reduction, and comorbidity management. Cardiorespiratory rehabilitation combined with weight loss has shown significant improvements in functional performance, exercise capacity, and muscle strength, especially when paired with non-invasive ventilation (NIV) ([28]).

Selecting the optimal therapy depends on the patient’s phenotype. CPAP is effective for obstructive events, but NIV, such as BiPAP-ST or BiPAP-AVAPS, is preferred when hypoventilation predominates, or CPAP fails to normalize CO_2_. NIV improves gas exchange, reduces PaCO_2_, and enhances arterial oxygenation, with benefits increasing proportionally with adherence ([60]).

Recent studies suggest that BiPAP-AVAPS may outperform CPAP in reducing PaCO_2_ and improving sleep architecture. However, while NIV reduces respiratory-related morbidity and mortality, its impact on cardiovascular outcomes is limited, and adherence remains a challenge ([38]; [54]).

Weight loss through diet or bariatric surgery can reverse many OHS-related complications, with bariatric surgery offering greater benefits due to the magnitude of weight reduction. Emerging therapies, such as incretin-based treatments, show promise in managing obesity-related complications, though their long-term efficacy in OHS remains uncertain ([17]; [80]).

Finally, it is important to mention that current evidence suggests that certain treatments for HF and SDB may have indirect benefits for OHS. Diuretics, by reducing pulmonary congestion and systemic fluid retention, can decrease upper airway edema, improving airflow and respiratory function during sleep ([19]; [71]). Beta-blockers and angiotensin-converting enzyme (ACE) inhibitors, by lowering sympathetic overactivity and enhancing cardiovascular function, may indirectly support better respiratory outcomes ([81]). PAP therapy, which is fundamental for managing OHS, not only alleviates hypercapnia and hypoxemia but also reduces cardiac preload and afterload, positively influencing heart failure outcomes ([41]; [43]; [75]). Additionally, newer agents such as SGLT2 inhibitors have demonstrated cardiopulmonary benefits, including improvements in left ventricular function and reductions in systemic inflammation, which may hold promise for a subset of patients with OHS and coexisting heart failure ([22]; [24]). These overlapping therapeutic strategies highlight the need for an integrated, multidisciplinary management approach.

### 6.1. Combining Physiological and Psychological Interventions

As seen in the previous paragraphs, patients with OHS, HF, and sleep dysfunction represent a vulnerable population with complex, overlapping physiological and psychological challenges. While extensive research has been conducted on the role of OSAS and obesity in neurocognitive impairments and psychiatric disorders, studies specifically addressing OHS are limited. Moreover, there is a notable lack of psychological protocols and non-pharmacological interventions dedicated to this group, whether for secondary prevention, treatment, or rehabilitation. Most available studies primarily focus on the effects of PAP therapy on medical, neurocognitive, and psychological outcomes. For instance, Baris et al. highlighted that cognitive dysfunction, depression, and anxiety are significant but under-recognized comorbidities in OHS. They found that short-term PAP therapy had positive effects on neurocognitive functions and psychological well-being, though further multicenter, long-term studies are necessary to establish its sustained benefits ([8]).

Beyond PAP therapy, addressing daytime drowsiness—a hallmark of OHS—is critical, as it contributes to physical function deficits, limiting patients’ ability to engage in regular exercise. This inactivity perpetuates oxidative stress associated with sleep disturbance, further compromising physical and mental health ([63]). Emerging evidence suggests that even modest improvements in physical fitness can bring about structural and functional changes in the brain, positively influencing cognitive function and overall well-being. This underscores the potential of integrating physiological and psychological interventions, such as tailored exercise programs, cognitive–behavioral therapy, and stress reduction techniques, into comprehensive care plans. Such an approach could enhance quality of life, bridge the existing treatment gap, and foster long-term rehabilitation for this underserved population.

### 6.2. Technology-Enabled Comprehensive Care Models

Digital therapies (DTx), an integral part of digital health, leverage high-quality software, advanced algorithms, and artificial intelligence (AI) to deliver therapeutic interventions that actively engage users, correcting maladaptive behaviors like low participation or treatment refusal—common in chronic conditions such as obesity, diabetes, and sleep disorders. Smartphone apps and wearable devices play a growing role in managing sleep-related conditions. Apps range from white-noise generators to advanced tools for tracking sleep patterns and ventilation adherence (e.g., ResMed MyAir™, Appnea-Q) ([82]). For positional obstructive sleep apnea (POSA), solutions like SomnoPose provide position monitoring and vibration alarms to promote corrective sleeping postures ([13]).

While consumer sleep technologies (CSTs) enhance patient–clinician interactions, their diagnostic and therapeutic validity is often limited, as highlighted by the AASM. Digital platforms, self-monitoring devices, and actigraphy tools—commonly worn on the wrist or ankle—offer data on sleep patterns, complementing clinical insights from polysomnography (PSG). Mobile technology further enables lifestyle interventions for obese patients with sleep apnea, with smartphone-based behavior change tools demonstrating promising results in weight loss and adherence improvement. However, the success of digital tools depends on usability, proper implementation, and their integration into clinical workflows. Patient-reported outcomes can bridge the gap between clinical data and patient experiences, enhancing the overall impact of digital health technologies ([45]; [72]).

## 7. Future Directions and Research Gaps

### 7.1. Unresolved Questions in Pathophysiology and Management

Telehealth and apps are appropriate strategies that could transform the approach to healthcare by providing increased and anytime access to information and empowering patients to be informed, engaged, and empowered to participate in shared decision making and effective self-management of chronic conditions ([51]), shifting healthcare from a disease-centered to a patient-centered model ([99]).

Several clinical trials have demonstrated the feasibility of telemedicine-based management of OSA and comorbidities compared to a more traditional face-to-face care model, suggesting non-inferiority in terms of CPAP treatment adherence, compliance ([34]), as well as functional outcomes such as satisfaction and cost-effectiveness ([35]).

Suarez-Giron and colleagues ([82]) found positive feedback with videoconferencing and mobile health (mHealth) interventions and implemented a telemedicine-based strategy for CPAP follow-up that was as effective as standard hospital-based care in terms of CPAP compliance and symptoms improvement, as well as lower cost ([35]).

Nevertheless, technological innovations in healthcare can be potentially detrimental if they are not properly applied or understood ([83]), a phenomenon previously described as the “technological labyrinth syndrome”, which basically refers to a process that is too complex for patients to follow or excessively time consuming for healthcare professionals to follow. Therefore, it seems imperative that any new technology used for health purposes is fundamentally simple, easy to use, reliable and transparent, and able to control or contact with the patient.

The use of telemedicine to enhance CPAP follow-up is an area that has been most explored, but the optimal format and information and delivery mechanism are still unclear ([33]).

Implementation of the technologies is an aspect that needs to be examined in detail, not only to create interesting and efficient digital health tools but also to ensure that participants have the support they need to integrate them into their daily lives and recognize their potential benefits for their health conditions. Larger studies need to be carried out to validate the clinical impact of digital interventions in patients with other chronic diseases while also meeting the respective implementation necessities.

Future studies should aim to analyze the effectiveness of different digital technologies that differ in design, user interface, and data collected, to find the most usable and patient-appropriate solution for the HF population that will increase patient engagement. In addition, it is essential to implement an effective way of communicating clinical data generated by digital health technologies to healthcare providers, which could lead to treatment adjustments or management decisions ([91]).

### 7.2. Integrating Digital Health in Research and Clinical Practice

Digital health innovations hold immense potential for addressing the unmet needs in research and clinical practice for patients with OHS, HF, and sleep dysfunction. Given the complexity of these conditions and their multifaceted physiological and psychological implications, DHIs can bridge gaps in care, enhance access to treatment, and drive personalized, patient-centered approaches.

In research, digital health technologies, such as wearable devices, telemonitoring systems, and advanced data analytics, enable continuous, real-time tracking of physiological parameters like oxygen saturation, transcutaneous carbon dioxide (TcCO_2_) levels, heart rate, and sleep patterns. These tools can facilitate large-scale data collection for longitudinal studies, offering valuable insights into disease progression, neurocognitive impacts, and psychological outcomes ([23]; [87]). For example, monitoring daytime drowsiness and physical activity levels via wearable devices can help identify patients at higher risk of heart failure and cognitive decline, tailoring interventions to their specific needs ([92]). Additionally, integrating patient-reported outcomes collected through digital platforms, such as mobile apps or telemedicine systems, allows researchers to evaluate the interplay of psychological distress, cognitive impairments, and physical symptoms over time.

In clinical practice, DHIs can transform the management of OHS, HF, and sleep dysfunction by improving early detection, treatment adherence, and patient engagement. Remote monitoring systems and teleconsultations enable continuous follow-up and timely intervention, particularly for patients with limited access to healthcare facilities. Digital tools like sleep trackers and PAP adherence monitoring systems can provide actionable feedback to both patients and healthcare providers ([84]; [53]), fostering adherence to treatment protocols and mitigating the psychological burden of chronic disease. Furthermore, DAs can support shared decision making, ensuring that patients’ preferences and health literacy levels are considered in treatment planning.

Despite our efforts to explore the psychosocial and cognitive dimensions of OHS, heart failure, and SDB, the current literature on their intersection remains sparse. This limitation prevented us from delving further into the shared psychosocial and cognitive effects of these interconnected conditions. Further studies are necessary to better understand these complex relationships. Combining DHIs with non-pharmacological interventions, such as cognitive–behavioral therapy (CBT) for managing anxiety and depression or digital platforms that deliver tailored exercise programs, could amplify therapeutic benefits. For instance, virtual coaching apps could guide patients in overcoming barriers to physical activity, mitigating oxidative stress, and enhancing cognitive and emotional well-being. Moreover, the integration of AI in digital platforms could facilitate the early identification of patients at risk for poorer outcomes, enabling personalized interventions (Figure 2).

Despite these opportunities, challenges remain, including ensuring equitable access to digital tools, addressing the digital literacy gap among older and less tech-savvy populations, and resolving interoperability issues with existing healthcare systems ([94]). Future efforts must focus on designing inclusive, user-friendly technologies and conducting multicenter trials to evaluate the efficacy of DHIs in improving long-term outcomes in these patient populations. By seamlessly integrating digital health into research and clinical care, we can address the complex needs of patients with OHS, HF, and sleep dysfunction, ultimately enhancing their quality of life and clinical outcomes.

#### Regulatory and Ethical Considerations in Digital Health Integration

The adoption of DHIs for managing respiratory and cardiovascular conditions, including OHS, HF, and SDB, presents both opportunities and challenges, particularly in terms of regulatory and ethical considerations. These frameworks, which govern the approval, safety, and implementation of digital tools such as telemedicine platforms, wearable devices, and mobile health applications, are evolving rapidly, especially after the COVID-19 pandemic ([39]). Regulatory requirements differ across regions, but consistent challenges include data security, patient privacy, and clinical validation of DTx ([37]; [64]; [86]). The distinct features of digital technology exacerbate pre-existing issues. Patients are empowered to actively participate in the co-maintenance of their own health as a result of the healthcare industry’s growing consumer-centric strategy. As a result, hospitals are gradually moving away from inpatient care and toward outpatient care, and the medical (tech) sector is increasingly interacting directly with patients, raising issues with data security and compliance. In this regard, biospecimens and data gathered for one reason could be used for completely other purposes later on without obtaining in their natural state consent. Furthermore, information that was initially collected in particular privacy situations could be used in more general, unexpected circumstances, potentially leading to user expectations being violated ([65]). Since technology businesses that operate in the healthcare industry sometimes do not come within the standard regulatory frameworks controlling scientific research, corporate involvement further complicates ethical monitoring. Concerns regarding “incidental research subjects”, or those who are inadvertently watched or examined just by virtue of their affiliation with data-sharing parties, are also brought up by DHIs ([64]). These concerns are made more difficult by emerging challenges. The sheer volume of data being collected, made possible by digital breakthroughs, presents issues with data privacy, quality, and the possibility of negative social effects like prejudice or stigma. The efficacy of data anonymization has been weakened by advances in computer science, increasing the risk of re-identification from purportedly de-identified datasets. Moreover, data science lacks established ethical frameworks, whereas biological research has a clear set of ethical standards. Stronger regulatory monitoring and ethical training are necessary in the quickly changing field of digital health, as digital technology enterprises frequently function independently with little external ethical standards ([37]; [64]; [86]). Clinicians and healthcare institutions must stay informed about these developments to ensure compliance and maximize patient benefits. Further research is needed to provide practical guidance on navigating these regulations and integrating compliant technologies effectively into clinical practice. Finally, it is relevant to consider that, to ensure the successful adoption of DHIs for managing OHS, HF, and SDB, cost-effectiveness is a critical consideration, as digital technologies may reduce healthcare expenses by optimizing disease management ([27]) and minimizing hospital readmissions ([18]). However, initial costs for implementation, training, and maintenance must be considered. Implementation guidelines should prioritize a phased approach, starting with pilot testing to evaluate feasibility and user experience. Comprehensive training for both healthcare providers and patients is essential to maximize the benefits of these technologies. Regular feedback and performance assessments should guide iterative improvements.

To support the effective integration of DHIs in clinical practice, we outline key recommendations across several dimensions, including technology selection, patient engagement, data security, ethical compliance, and implementation strategies. These guidelines aim to assist clinicians, healthcare organizations, and patients in responsibly adopting and evaluating digital tools to enhance care for conditions such as OHS, HF, and SDB. A detailed summary is provided in Table 3.

### 7.3. Limitations

As this is a narrative review, we aimed to provide a broad and comprehensive synthesis of the existing literature on OHS, HF, and SDB, focusing on their pathophysiology, psychosocial dimensions, digital health solutions, and therapeutic interventions. Given the diversity of study designs, clinical settings, and outcome measures in this area, it is challenging to capture every single relevant publication comprehensively. Given the diverse nature of these areas, the evidence base varies in terms of rigor, robustness, and availability. In sections that focus on the pathophysiological mechanisms of OHS, HF, and SDB, the evidence is relatively well established, with numerous studies supporting the understanding of the physiological and systemic interactions. However, in areas such as the integration of digital health solutions and their effectiveness in managing these conditions, the evidence is still evolving. Many studies in this domain are in early stages, often with smaller sample sizes or pilot designs, and more robust evidence is needed to establish conclusive clinical recommendations.

We recognize that this variability in evidence quality may influence the interpretation and application of the findings. In particular, the lack of high-quality data in certain areas, such as digital health interventions, may limit the ability to make strong, evidence-based recommendations for their widespread adoption in clinical practice. We have revised the manuscript to clearly highlight these discrepancies, providing a more transparent view of the current evidence landscape. We also discuss the implications of this variability, emphasizing that while certain treatment modalities have well-supported clinical evidence, others, particularly in the realm of digital health, warrant further investigation and validation.

As this is a narrative review, it is important to note that varying levels of evidence may be present across different sections, and a qualitative rating of the studies was not deemed necessary for the purpose of synthesizing the current understanding and providing a broad perspective on the topic. We acknowledge the potential for selection bias in the literature coverage, as our inclusion criteria were designed to select key studies that provide significant insights into the intersection of OHS, HF, and SDB. We focused on studies that addressed both physiological and psychological aspects, with the aim of presenting a holistic view of the current evidence. While we sought to include a range of study types (including clinical trials, observational studies, and expert opinion papers), the non-systematic nature of this narrative review means that not all relevant studies could be included. To minimize potential bias, we carefully reviewed the most relevant sources, considering a balance of clinical research, interventions, and emerging areas such as digital health solutions. However, we recognize that the focus of this review may have unintentionally excluded some studies, particularly those with narrow or specialized focuses. We have made efforts to clearly describe our approach to study selection and provide a balanced overview of the existing literature. This narrative review aims to synthesize existing knowledge, with the understanding that future systematic reviews may address gaps in coverage and further refine conclusions.

## 8. Conclusions

In summary, while NIV, particularly PAP therapy, has demonstrated short-term benefits in improving respiratory and neurocognitive functions, depression, and anxiety in patients with OHS, its long-term effects remain underexplored. The available evidence highlights key challenges, such as the persistence of residual AHI, reduced physical activity, and ongoing impairments in quality of life, despite significant adherence to ventilation therapy. These findings suggest that factors beyond adherence, such as hypercapnia and daytime respiratory symptoms, may play a critical role in influencing patient outcomes.

Future efforts must focus on comprehensive approaches that integrate patient education, personalized optimization of NIV settings, and strategies to address residual respiratory events. Enhancing patient education to emphasize not just adherence but also the quality and effectiveness of NIV usage is paramount. Moreover, larger multicenter prospective studies are needed to identify the factors contributing to reduced physical activity and quality of life, as well as to evaluate the long-term impact of PAP therapy on both physiological and psychological outcomes.

Finally, the integration of digital health technologies into research and clinical practice offers a promising avenue to address these challenges. By combining physiological and psychological interventions with digital tools for monitoring, feedback, and education, we can move toward more holistic and effective care strategies that improve outcomes and quality of life for patients with OHS and other complex sleep-disordered breathing conditions.

## Figures and Tables

**Figure 1 behavsci-15-00285-f001:**
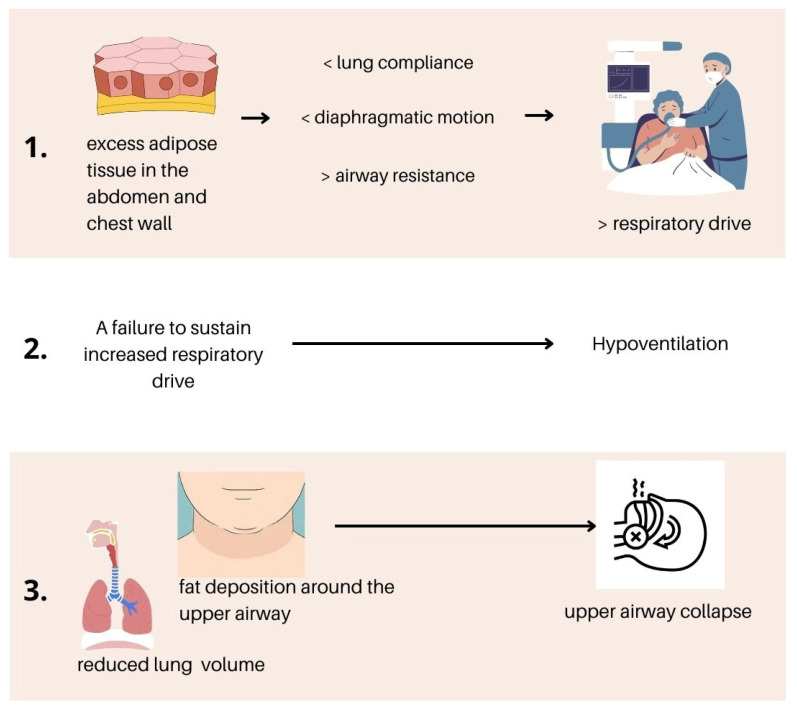
Pathophysiological mechanisms underlying OHS determining a change in cardiac functioning.

**Figure 2 behavsci-15-00285-f002:**
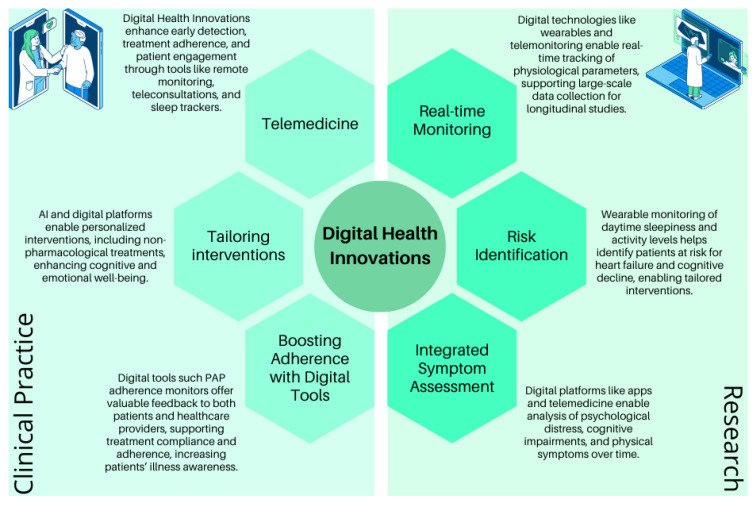
Digital Health’s implementation in the OHS.

**Table 1 behavsci-15-00285-t001:** Inclusion and exclusion criteria.

	Inclusion Criteria	Exclusion Criteria
**Study Design**	Peer-reviewed studies, including clinical trials, observational studies, cohort studies, qualitative studies, and case reports.	Studies not published in peer-reviewed journals (e.g., conference abstracts, non-peer-reviewed articles); grey literature
	Literature reviews and meta-analyses that provide insight into the topic	Animal studies or preclinical studies not directly applicable to human health.
		Studies focusing solely on conditions unrelated to OHS, HF, or SDB, such as purely obstructive lung diseases (e.g., asthma or COPD) without sleep-related components.
**Language**	English	Studies published in languages other than English
**Study focus**	Research addressing Obesity Hypoventilation Syndrome (OHS), Heart Failure (HF), and Sleep Disordered Breathing (SDB), particularly studies exploring their pathophysiology, clinical management, psychosocial aspects, and therapeutic approaches in digital terms	Studies that do not directly address OHS, HF, or SDB, or do not explore the intersection between these conditions.
	Studies that focus on the intersection between OHS, HF, and SDB, exploring both physiological and psychological, cognitive and behavioral dimensions	Articles focused on unrelated health conditions or diseases, even if they involve similar physiological or psychological mechanisms.
	Research on digital health interventions, including telemedicine, remote monitoring, and digital therapeutic approaches in the context of OHS, HF, and SDB.	Research that only addresses one condition in isolation (e.g., studies focusing only on HF without consideration of OHS or SDB).
**Outcomes**	Studies that report on clinical, psychological, or physiological outcomes, or those that discuss mechanisms linking OHS, HF, and SDB.	Studies that do not report on outcomes relevant to the understanding of OHS, HF, or SDB, or that focus exclusively on unrelated health outcomes.
	Research that assesses the efficacy, feasibility, or impact of digital health interventions on these conditions.	Articles without sufficient data or empirical evidence on clinical, psychological, or therapeutic outcomes.
**Population**	Studies involving adults diagnosed with OHS, HF, or SDB (including obstructive sleep apnoea and related disorders).	Studies focusing on populations outside of adults (e.g., children or elderly patients not typical of the population under consideration).
	Research focusing on both general and clinical populations (e.g., hospital settings, rehabilitation centers).	Research that only examines a population with a specific co-morbidity not relevant to the review (e.g., studies focusing solely on diabetes without addressing OHS, HF, or SDB).

**Table 2 behavsci-15-00285-t002:** Characteristics of the considered studies.

Author, Year	Methodology	DiscussedPathology	Main Topics	Key Points	Main Results
[3] ([3])	Prospective Observational Study	Obesity Hypoventilation Syndrome (OHS), Pulmonary Hypertension	Prevalence of Pulmonary Hypertension in OHS	Investigates the prevalence of pulmonary hypertension in OHS patients, providing insights into its cardiovascular impact and the necessity for early diagnosis.	Identified various smartphone applications that enhance OSA management. Found that while digital solutions improve accessibility, their effectiveness varies based on user compliance and device accuracy.
[17] ([17])	Prospective Observational Study	Obesity Hypoventilation Syndrome (OHS), Sleep Disorders	Adherence to Positive Airway Pressure (PAP) Therapy and Long-Term Outcomes	Examines the association between adherence to PAP therapy and long-term health outcomes in OHS patients, emphasizing the importance of compliance for improved quality of life and reduced complications.	Demonstrated that digital health interventions can be cost-effective by reducing healthcare utilization and improving patient outcomes in chronic disease management.
[8] ([8])	Observational Study	Obesity Hypoventilation Syndrome (OHS), Sleep Disorders	Effect of PAP Therapy on Neurocognitive Functions, Depression, and Anxiety in OHS	Evaluates the impact of PAP therapy on cognitive function and psychological well-being in OHS patients, supporting its role in enhancing mental health outcomes.	Showed that telemonitoring significantly improved CPAP adherence in sleep apnea patients compared to standard care. Patients using telemedicine solutions had better compliance and health outcomes.
[13] ([13])	Systematic review	Obstructive Sleep Apnea	Smartphone applications for OSA	Discusses various smartphone applications for detecting and managing OSA.	Provided an updated overview of mobile technologies for OSA management, highlighting their effectiveness in monitoring and treatment. It also discusses the advantages (usability, effectiveness, and technological advancements) and limitations of current digital health solutions.
[27] ([27])	Systematic review	Digital health interventions	Cost-effectiveness of digital health	Examines the economic feasibility and effectiveness of digital health interventions. The review highlights how integrating these technologies can enhance chronic disease management while reducing healthcare costs.	Provided strong evidence that telemonitoring enhances CPAP adherence and improves clinical outcomes in sleep apnea patients, validating the role of digital health interventions.
[34] ([34])	Randomized Controlled Trial	Sleep Disorders, Digital Health	Telemonitoring and CPAP	Investigates the impact of telemedicine education and telemonitoring on CPAP adherence in sleep apnea patients, demonstrating improved compliance and treatment efficacy through digital health interventions.	Revealed that digital home monitoring reduced hospitalizations and improved self-management in heart failure patients. Digital health solutions were particularly effective in remote care settings.
[54] ([54])	Randomized Controlled Trial	Obesity Hypoventilation Syndrome	Long-Term Noninvasive Ventilation in OHS Patients Without Severe OSA	Evaluates the impact of long- term noninvasive ventilation on OHS patients, providing evidence for its role in improving respiratory function and overall health outcomes.	Identified key barriers (e.g., lack of digital literacy, cost concerns) and facilitators (e.g., improved remote monitoring, patient empowerment) for adopting digital health in cardiovascular care.
[72] ([72])	Randomized Controlled Trial (RCT)	Heart Failure, Digital Health	Home Monitoring with Digital Technology in Chronic Heart Failure	Assesses the effectiveness of digital home monitoring for heart failure patients, highlighting improvements in disease management and potential reductions in hospitalizations.	Indicated that mobile health applications positively influenced CPAP adherence and patient engagement. Patients using digital tools demonstrated higher treatment compliance.
[82] ([82])	Observational Study	Sleep Disorders, Digital Health Intervention	Mobile Health App for CPAP Therapy Support	Assesses the feasibility and effectiveness of a mobile health application designed to support CPAP therapy adherence in sleep apnea patients, highlighting the benefits of digital interventions in patient engagement.	Showed that long-term noninvasive ventilation in OHS patients improved respiratory outcomes, supporting its role in disease management.
[94] ([94])	Systematic scoping review	Digital health in cardiovascular care	Adoption of digital health in cardiology	Highlights barriers and opportunities in adopting digital health in cardiology. This review emphasizes the potential of digital health technologies to enhance cardiovascular disease management.	Identified critical barriers (e.g., lack of digital literacy, infrastructure challenges) and facilitators (e.g., improved monitoring, patient empowerment) for adopting digital health in cardiology.

**Table 3 behavsci-15-00285-t003:** Practical Guidelines for Technology Adoption in Clinical Practice.

	Recommendations for Clinicians and Healthcare Professionals	Recommendations for Patients
**Technology Selection**	Choose digital solutions that have demonstrated clinical effectiveness and usability through peer- reviewed evidence or, at least, pilot studies. Evaluate certifications and compliance with medical device standards.	Verify that the app or tool is recommended by your healthcare provider or a reputable source. Look for user reviews and check for official certifications.
**Patient Engagement**	Ensure patients are educated on how to use digital health tools effectively, emphasizing user-friendly interfaces and patient-centered features. Encourage patient feedback.	Ask for clear guidance on how to use digital health tools and seek help if confused. Ensure the app is easy to use and helpful for your care needs.
**Data Security and Privacy**	Adopt technologies compliant with data protection regulations (e.g., GDPR, HIPAA). Conduct regular data security audits.	Check privacy policies to see how your data are handled. Avoid sharing sensitive data on unsecured platforms.
**Ethical Compliance**	Ensure digital solutions have transparent data usage policies. Obtain informed consent before digital data collection.	Be aware of what data are being collected and why. Give consent only if you feel informed and comfortable.
**Implementation Strategies**	Pilot the adoption of technologies on a smaller scale before scaling up across clinical settings. Monitor for technical and clinical issues.	Start by using digital tools gradually. Report any problems or issues to your healthcare provider.
**Outcome Evaluation**	Develop standardized metrics for assessing the effectiveness and impact of digital health tools on patient outcomes and clinical workflows.	Keep track of how digital tools affect your health and report changes or concerns.

## Data Availability

The authors confirm that the data supporting the findings of this study are available within the article.

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
