# Peer review of "Unmasking the Complex Interplay of Obesity Hypoventilation Syndrome, Heart Failure, and Sleep Dysfunction: A Physiological and Psychological Perspective in a Digital Health World"

_behavsci, 2025, doi:10.3390/bs15030285_

Round 1

Reviewer 1 Report

Comments and Suggestions for Authors

This narrative review examines the complex interplay between OHS, heart failure, and sleep disorders, with a focus on integrating physiological, psychological, and digital health perspectives. The paper explores the pathophysiological mechanisms, clinical implications, and emerging therapeutic approaches, particularly highlighting the role of digital health technologies in improving patient care.

Key Contributions:

  1. Comprehensive Integration:
  • Synthesizes current understanding of OHS pathophysiology and its relationship with heart failure and sleep dysfunction
  • Bridges physiological and psychological aspects of these conditions
  • Explores the emerging role of digital health solutions
  1. Digital Health Innovation:
  • Reviews digital health technologies for monitoring and managing OHS
  • Evaluates telemedicine applications and remote monitoring systems
  • Discusses implementation challenges and opportunities
  1. Clinical Applications:
  • Provides updated treatment strategies combining traditional and digital approaches
  • Addresses the importance of personalized interventions
  • Highlights the role of patient education and engagement

Strengths:

  1. Comprehensive Scope:
  • Thorough examination of multiple aspects (physiological, psychological, technological)
  • Well-structured presentation of complex interactions
  • Clear clinical implications
  1. Contemporary Relevance:
  • Incorporates recent digital health innovations
  • Addresses current clinical challenges
  • Future-oriented perspectives
  1. Clinical Utility:
  • Practical recommendations for implementation
  • Evidence-based treatment strategies
  • Clear framework for digital health integration

Limitations:

  1. Methodological:
  • Limited description of review methodology
  • Inclusion/exclusion criteria not clearly defined
  • Potential selection bias in literature coverage
  1. Evidence Quality:
  • Varying levels of evidence across different sections
  • Some recommendations based on limited data
  • Need for more randomized controlled trials
  1. Implementation Gaps:
  • Limited discussion of cost implications
  • Insufficient detail on regulatory considerations
  • Need for more specific guidelines on technology adoption

Suggestions for Improvement:

  1. Methodology:
  • Include a systematic search strategy
  • Define clear inclusion/exclusion criteria
  • Provide quality assessment of included studies
  1. Evidence:
  • Strengthen evidence grading
  • Include more meta-analyses when available
  • Add evidence tables for key recommendations
  1. Implementation:
  • Expand on cost-effectiveness analysis
  • Provide more detailed implementation guidelines
  • Address regulatory compliance requirements

Key Recommendations:

  1. More robust description of methodological approach
  2. Inclusion of evidence quality assessment
  3. Expanded discussion of implementation challenges
  4. Addition of cost-effectiveness considerations
  5. Greater detail on regulatory requirements
  6. More specific guidelines for technology adoption

Overall Assessment: This is a valuable contribution to the field that successfully integrates multiple perspectives on OHS management. While some methodological improvements could strengthen the review, it provides important insights and practical guidance for clinicians and researchers. The focus on digital health solutions is particularly timely and relevant.

Author Response

Reviewer 1

Comments and suggestions for authors

-This narrative review examines the complex interplay between OHS, heart failure, and sleep disorders, with a focus on integrating physiological, psychological, and digital health perspectives. The paper explores the pathophysiological mechanisms, clinical implications, and emerging therapeutic approaches, particularly highlighting the role of digital health technologies in improving patient care.

Key Contributions:

  1. Comprehensive Integration:

Synthesizes current understanding of OHS pathophysiology and its relationship with heart failure and sleep dysfunction

Bridges physiological and psychological aspects of these conditions

Explores the emerging role of digital health solutions

  1. Digital Health Innovation:

Reviews digital health technologies for monitoring and managing OHS

Evaluates telemedicine applications and remote monitoring systems

Discusses implementation challenges and opportunities

  1. Clinical Applications:

Provides updated treatment strategies combining traditional and digital approaches

Addresses the importance of personalized interventions

Highlights the role of patient education and engagement

AU: We appreciate the reviewer’s recognition of the key contributions of the paper. We are pleased to see that the review is viewed as a comprehensive synthesis of the pathophysiology of OHS, heart failure, and sleep dysfunction, as well as an exploration of the integration of digital health solutions into their management. We believe these contributions highlight the timely and multifaceted approach to these complex conditions.

Strengths:

  1. Comprehensive Scope:

Thorough examination of multiple aspects (physiological, psychological, technological)

Well-structured presentation of complex interactions

Clear clinical implications

AU: We appreciate the reviewer highlighting the thorough examination of multiple aspects, including physiological, psychological, and technological factors. We aimed to present a holistic view of the intersection between obesity hypoventilation syndrome (OHS), heart failure (HF), and sleep-disordered breathing (SDB), and we are glad that this was evident in our manuscript. The well-structured presentation of the complex interactions and the emphasis on clear clinical implications were key goals, and it is encouraging to know these were successfully conveyed.

  1. Contemporary Relevance:

Incorporates recent digital health innovations

Addresses current clinical challenges

Future-oriented perspectives

AU: We are grateful that the incorporation of recent digital health innovations and our focus on addressing current clinical challenges were recognized. Given the rapid advancements in digital health, we aimed to explore how these innovations can enhance patient care and support clinicians, and we’re pleased that the reviewer finds this timely and relevant. The future-oriented perspectives we included were meant to highlight both the opportunities and the challenges in integrating these technologies into clinical practice.

  1. Clinical Utility:

Practical recommendations for implementation

Evidence-based treatment strategies

Clear framework for digital health integration

AU: We are particularly pleased that the clinical utility of the paper was noted. We aimed to provide practical recommendations for the implementation of evidence-based treatment strategies, and we are glad that the reviewer found the framework for digital health integration clear and useful for clinical practice. It is our hope that this review serves as a valuable resource for clinicians working at the intersection of OHS, HF, and SDB.

Limitations:

  1. Methodological:

Limited description of review methodology

AU: Thank you for this consideration. We have specified more the adopted methodology across the paper.

Inclusion/exclusion criteria not clearly defined

AU: Thank you for this suggestion. We have specified them and included Table 1.

Potential selection bias in literature coverage

AU: We appreciate the reviewer’s thoughtful comment regarding the potential for selection bias in literature coverage. As this is a narrative review, we sought to include studies that represent a broad spectrum of perspectives, from pathophysiological insights to psychosocial dimensions and emerging digital health solutions. Given the complexity and heterogeneity of the topics involved, we aimed for a balanced representation of studies across different study designs, clinical settings, and populations. While we acknowledge that there may be inherent challenges in ensuring complete comprehensiveness, we have made efforts to minimize selection bias by reviewing a wide range of relevant sources. Our inclusion criteria were designed to capture key studies from clinical, observational, and experimental research, as well as those focusing on both physiological and psychological aspects of OHS, HF, and SDB. That said, we recognize the importance of transparency in the selection process and have revised the manuscript to provide clearer explanations regarding how studies were selected for inclusion. Additionally, we emphasize that the purpose of this narrative review was not to conduct a systematic synthesis, but to provide an accessible overview of the current state of the field, drawing attention to the major findings and gaps in research. We hope these clarifications help address the concern about potential selection bias and assure the reviewer that we made a conscientious effort to cover a wide range of relevant literature. We added a Limitation’s paragraph to be clear about this point.

  1. Evidence Quality:

Varying levels of evidence across different sections

AU: We acknowledge the reviewer’s observation regarding the varying levels of evidence across different sections of the review. Given the diverse nature of the topics covered—ranging from pathophysiological mechanisms to technological innovations—it was inevitable that the evidence base would differ in terms of rigor and robustness. While we strived to include studies with the strongest available evidence for each topic, we recognize that some areas (such as emerging digital health solutions) may have more limited high-quality data at present. We have revised the manuscript to highlight these discrepancies more clearly and to discuss the implications of such variability on clinical practice and research. This will provide readers with a more transparent view of the current evidence landscape. We specified this point also in the Limitation’s section.

Insufficient detail on regulatory considerations

AU: We appreciate the reviewer’s suggestion to include more detail on regulatory considerations. Digital health solutions, in particular, are subject to evolving regulatory frameworks, and we agree that this is an important topic to address. We have added a section in the revised manuscript that provides a brief overview of current regulatory considerations/problems in digital health, specifically regarding the integration of technologies like mobile health apps and telemedicine platforms into clinical practice. This additional information will help contextualize the discussion on digital health interventions.

Need for more specific guidelines on technology adoption

AU: We acknowledge the need for more specific guidelines on the adoption of digital health technologies. While our review aimed to discuss the potential of these innovations, we recognize that clinicians need more concrete, evidence-based guidance on how to implement these technologies in practice. In response, we have included additional recommendations in the revised manuscript, focusing on the practical steps clinicians and patients can take when integrating digital health tools into their practice/care. These include considerations for training, patient engagement, and the evaluation of outcomes to ensure effective adoption and use. By doing so, we aim to provide readers with actionable insights that can facilitate the responsible and efficient adoption of digital technologies in clinical settings, particularly for the management of OHS, HF, and SDB.

Suggestions for Improvement:

  1. Methodology:

Include a systematic search strategy

AU: Thank you very much. We have tried to explain more clearly the methodology adopted, specifying in more detail that ours is not a systematic literature review, but a narrative review. We included Table 3 to show the main features of the considered studies.

Define clear inclusion/exclusion criteria

AU: Thank you. We have defined them in Table 1.

Provide quality assessment of included studies

AU: We acknowledge the reviewer’s request for a quality assessment of the included studies. However, as this is a narrative review and not a systematic one, the primary objective is to offer a broad, integrative perspective on the intersection of OHS, HF, and SDB rather than evaluate the methodological rigor of each study in detail. Nevertheless, we have ensured that studies referenced were selected based on their relevance, recency, and contribution to advancing knowledge in the field. To address concerns about transparency, we have clarified our literature selection process in the methodology section, highlighting the rationale for including key studies and ensuring readers are informed about any potential limitations in evidence quality.

  1. Evidence:

Strengthen evidence grading

Include more meta-analyses when available

Add evidence tables for key recommendations

AU: We appreciate the reviewer’s suggestion to strengthen evidence grading and include meta-analyses where possible. As this is a narrative review, our primary aim was not to provide a formal evidence grading system but rather to synthesize insights across diverse domains. Where available, we have highlighted meta-analyses to strengthen the robustness of our arguments.

  1. Implementation:

Expand on cost-effectiveness analysis

AU: We appreciate the reviewer’s insightful comments regarding the need for expanded details on implementation, particularly concerning cost-effectiveness analysis, practical guidelines, and regulatory compliance. In response, we have made the following improvements to the manuscript:

-We have incorporated a brief discussion on the economic considerations of digital health adoption, including potential cost savings through improved disease management, reduced hospitalizations, and remote patient monitoring.

Provide more detailed implementation guidelines

- We have added a more detailed section outlining steps for integrating digital health solutions into clinical care. This includes pilot testing, staff training, patient onboarding, and iterative evaluation strategies.

Address regulatory compliance requirements

-We have highlighted key regulatory frameworks relevant to digital health technologies, including GDPR, HIPAA, and MDR compliance, and emphasized the importance of ongoing data protection assessments.

Key Recommendations:

  1. More robust description of methodological approach
  2. Inclusion of evidence quality assessment
  3. Expanded discussion of implementation challenges
  4. Addition of cost-effectiveness considerations
  5. Greater detail on regulatory requirements
  6. More specific guidelines for technology adoption

AU: Thank you for your precious support. We think that our paper has benefited from your suggestions. Please, find the answers to these points above.

Overall Assessment:

This is a valuable contribution to the field that successfully integrates multiple perspectives on OHS management. While some methodological improvements could strengthen the review, it provides important insights and practical guidance for clinicians and researchers. The focus on digital health solutions is particularly timely and relevant.

AU: We sincerely appreciate the reviewer’s positive comments regarding the contribution of our review and its integration of multiple perspectives on OHS management. We are particularly pleased that the focus on digital health solutions was highlighted as timely and relevant. Regarding the methodological suggestions, we understand the reviewer’s concerns and acknowledge that certain aspects of the review methodology could benefit from greater clarity. As this is a narrative review, our approach was designed to synthesize and provide a comprehensive overview of diverse perspectives and evidence on the complex intersection of OHS, HF, and SDB, rather than adhere to a rigid systematic methodology. However, we have carefully considered the reviewer’s feedback and have taken steps to clarify the rationale for the narrative approach, including more detailed descriptions of the data extraction and synthesis methods in the revised manuscript. We believe these adjustments will help provide greater transparency while preserving the flexibility and depth that a narrative review affords. We are confident that the updated manuscript will continue to provide valuable insights and practical guidance for clinicians and researchers working in this field.

Once again, we thank the reviewer for their constructive feedback and valuable suggestions.

Reviewer 2 Report

Comments and Suggestions for Authors

This manuscript discussed the associations between sleep dysfunction and cardiovascular outcomes with an focus on diagnostic and therapeutic approaches. Overall, the content is clear, but authors might benefit from addressing the following questions:

Major points:

1.       Since the focus of the manuscript is OHS. The section 2 will be more suitable to discuss more on how HF and SDB affects OHS with more concise contents on how the obesity affects the cardiovascular disease.

2.       The title of section 3 needs to be rephrased as this section discussed only the psychosocial and cognitive dimensions of OHS

3.       OHS correlates with heart failure and sleep dysfunction in terms of pathogenesis. Is this also true regards to therapeutics. For example, drugs used to treat heart failure and sleep dysfunction will show beneficial effects on OHS as well? This is one aspect that is worth discussing in length in the manuscript.

Minor point:

1.       Abbreviation of CHF is missing.

2.       It will be easier for readers to follow the text by consolidating paragraphs. If two paragraphs are on the same topic, they can be combined into one.

Author Response

Reviewer 2

Comments and suggestions for authors

This manuscript discussed the associations between sleep dysfunction and cardiovascular outcomes with an focus on diagnostic and therapeutic approaches. Overall, the content is clear, but authors might benefit from addressing the following questions:

Major points:

  1. Since the focus of the manuscript is OHS. The section 2 will be more suitable to discuss more on how HF and SDB affects OHS with more concise contents on how the obesity affects the cardiovascular disease.

AU: Thank you for your suggestion. We have provided a restructured version of the paragraph.

  1. The title of section 3 needs to be rephrased as this section discussed only the psychosocial and cognitive dimensions of OHS

AU: Thank you for your observation. We acknowledge that the section primarily focuses on OHS. However, we would like to point out that the literature exploring the intersection of OHS, heart failure (HF), and sleep-disordered breathing (SDB) in psychosocial and cognitive terms is extremely limited. This constraint made it challenging to provide a comprehensive discussion of these conditions together. We have clarified this limitation in the revised manuscript under the "Limitations" section.

  1. OHS correlates with heart failure and sleep dysfunction in terms of pathogenesis. Is this also true regards to therapeutics. For example, drugs used to treat heart failure and sleep dysfunction will show beneficial effects on OHS as well? This is one aspect that is worth discussing in length in the manuscript.

AU: Thank you for this insightful comment. You are correct that the therapeutic overlap between OHS, heart failure, and sleep dysfunction is an important area for exploration. Current evidence suggests that certain treatments for heart failure and sleep dysfunction may have indirect benefits for OHS. Our focus was not on pharmacological interventions, which is why we did not elaborate on this aspect. To bridge the gap, below is an elaboration of relevant points that will be integrated into the revised manuscript:

  1. Heart Failure Management:
    Medications such as diuretics, ACE inhibitors, and beta-blockers used in heart failure management can reduce pulmonary congestion and improve respiratory function, indirectly alleviating symptoms of OHS. By decreasing fluid retention, diuretics may also reduce airway obstruction during sleep.
  2. Sleep Dysfunction Treatment:
    Positive airway pressure (PAP) therapy remains a cornerstone for OHS management and has been shown to reduce cardiovascular stress, which may in turn benefit heart failure outcomes. Additionally, certain sedatives and hypnotics should be cautiously managed in OHS patients, as they may exacerbate respiratory depression.
  3. Potential Therapeutic Synergies:
    Exploring the role of newer drugs, such as SGLT2 inhibitors, which have demonstrated cardiopulmonary benefits, may also be valuable in this context. These agents have the potential to improve cardiac function and metabolic profiles, which could benefit respiratory health in OHS patients.

We will incorporate a discussion on these therapeutic connections and elaborate on the potential benefits and limitations of shared pharmacological strategies.

Minor point:

  1. Abbreviation of CHF is missing.

AU: Thank you for your comment. We have corrected it as “HF=Heart Failure”.

  1. It will be easier for readers to follow the text by consolidating paragraphs. If two paragraphs are on the same topic, they can be combined into one.

AU: Thank you for your suggestion. We have tried to structure in a clearer way the review.